# Nature-Based Solutions in "Forest–Wetland" Spatial Planning Strategies to Promote Sustainable City Development in Tianjin, China

**Yangli Li** [1,2] , **Gaoyuan Wang** [2] , **Tian Chen** [2,*] , **Rui Zhang** [2,3] , **Long Zhou** [2,4] and **Li Yan** [1]

[1] School of Civil Engineering and Architecture, Southwest University of Science and Technology, Mianyang 621010, China; liyangli@tju.edu.cn (Y.L.); anniey007@cqu.edu.cn (L.Y.)

[2] School of Architecture, Tianjin University, Tianjin 300072, China; 1020206035@tju.edu.cn (G.W.); zhangrui1017_@tju.edu.cn (R.Z.); lzhou@cityu.mo (L.Z.)

[3] College of Design and Engineering, National University of Singapore, Singapore 117575, Singapore

[4] Faculty of Innovation and Design, City University of Macau, Macau 999078, China

[*] Correspondence: teec@tju.edu.cn

**Abstract:** Nature-based solutions are some of the most effective strategies to promote sustainable city development; however, existing research on NbS is mostly comprised of single variable studies rather than multiple variables. The purpose of this study was to explore the possibility of extending the NbS of a single variable to two variables for the better development of sustainable cities. Both forestation and wetland restoration are regarded as NbS for sustainable city development. The research approach of "forest–wetland" NbS was proposed and centers on the process and core issues of traditional NbS. Taking Tianjin as an example, the spatial patterns of forests and wetlands, correlation between the spatial distribution of forests and wetlands, and spatial correlation between the areas of forest growth and wetland growth within a certain distance in different years were studied using a spatial distribution pattern analysis, geographic concentration analysis, kernel density estimation and spatial autocorrelation analysis. Based on the core issues of NbS and the above spatial analysis, a "forest–wetland" spatial planning strategy was formulated. The main conclusions are as follows: forest and wetland were negatively correlated in the whole area of Tianjin, forest resources w mainly located in north, while wetland resources were mainly located in south. Compared with forests, the spatial distribution of wetlands in Tianjin was more balanced. There exist synergy and trade-offs between forest and wetland area under certain circumstances. Growth of forests was positively correlated with the growth of wetlands, within a distance of 0–400 m from 2000 to 2010, and within a distance of 0–600 m from 2010 to 2020. An increase in forest area will lead to an increase in evaporation, which in turn will hinder the growth of wetlands in Tianjin. Forest–wetland ecological network could promote synergistic between forest and wetland, and grey infrastructure to reduce potential trade-off between forest and wetland.

**Keywords:** nature-based solutions; forest; wetland; spatial planning strategies; sustainable city development

## 1. Introduction

Cities are facing challenges such as climate change, ecosystem destruction, and impeded sustainable city development. Researchers have proposed solutions based on nature to address such challenges in an ecologically adaptive manner. One of the most effective solutions currently available is the nature-based solution (NbS) [1,2]. Biomimicry is a prototype for NbS applications [3]. During the late 2000s, the term "natural-based solutions" emerged [2], and was mentioned in an official document issued by the World Bank in 2008 [4]. The International Union for Conservation of Nature (IUCN) defined NbS as "actions to protect, sustainably manage, and restore natural or modified ecosystems, that

address societal challenges effectively and adaptively, simultaneously providing human well-being and biodiversity benefits." [2]. The European Commission (EC) defined NbS as "actions inspired and supported by nature, which are cost-effective, simultaneously provide environmental, social, and economic benefits and help build resilience." [5]. The IUCN and EC definitions of NbS represent different perspectives. The IUCN's definition of NbS aims at ecologically sustainable development, whereas the EC's definition of NbS integrates social, economic, and environmental co-development. EC extended the definition of NbS based on that of the IUCN, which was reflected in the reconstruction of an opposing relationship between society and ecology and socio-ecology vantage point. This also set the foundation for NbS research of follow-up researchers who focused on the interrelationship between humans and nature. Sowińska-Świerkosz and Garcia (2022) proposed the ideas that the integration of blue-green elements is a key criterion of NbS, and hybrid green-blue-grey intervention can also be regarded as NbS to achieve enhanced efficiency and affordability [6]. Ma et al. (2022) mentioned that NbS should mitigate climate change, enhance the ecological environment, and protect biodiversity to achieve a win–win scenario [7]. NbS has been under development for decades; however, its theoretical underpinnings require further refinement. Scholars have further contributed to the core issues and holistic framework of NbS [6,8] to assist future NbS research advancement according to a defined set of processes (Figure 1).

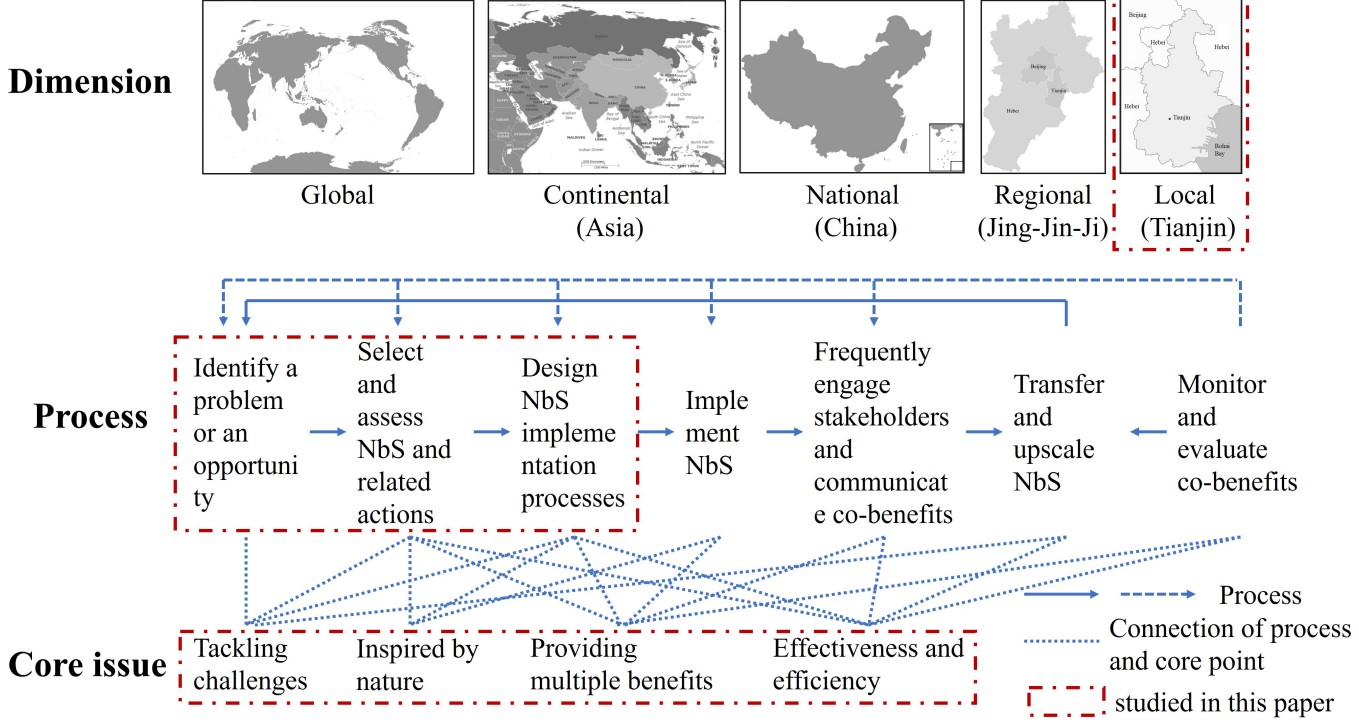

**Figure 1.** Process and core issues of NbS.

The primary challenge for NbS is climate change followed by sustainable city development [6]. To address the challenges of climate change, researchers explored hydro-climatic risks [9], urban heat island mitigation [10], carbon sequestration [11], and flood risk reduction [12]. As for the challenges of sustainable urban development, researchers have focused on shaping blue-green spaces, such as optimising urban parks, forests, and wetlands, to strengthen public health [13], raise incomes [7], and improve the quality of urban ecosystems [14]. Among these, NbS studies on forestation and wetland restoration require further exploration. The primary purpose of forestation within the context of NbS is to increase the extent of a carbon sink. Secondly, biodiversity can be improved through interplanting and economic benefits can be promoted by economic forests. Wetland restoration within NbS

can be adopted to reduce the level of water pollutants and upgrade the downstream water quality, since wetlands act as natural filtration systems. As NbS evolves into a comprehensive solution, the synergy and trade-off between its systems cannot be ignored [6,15,16]. An interactive relationship exists between wetlands and forests. Wetlands have the capacity for water storage [17] and can provide sufficient water for forest growth. However, the increase in forests causes greater transpiration and reduces the total water supplied to the wetlands, resulting in loss of wetlands. This forest-induced reduction in wetlands is particularly significant in northern and northeastern China [15].

The application of NbS requires an understanding of its interactions with the surrounding environment at different scales [18]; this supports the need for studying the relationship between two or more variables in NbS. For forests and wetlands, existing studies have only considered the advantages that can emerge from the management of these two ecological resources; however, these studies did not integrate the synergy (the increase of forest/wetland causes the increase of wetland/forest) and trade-offs (the increase of forest/wetland causes a decrease in wetland/forest) between them. The purpose of this study was to explore the possibility of extending the NbS of a single variable to two variables. This study used Tianjin as a study site to analyse the spatial relationship between forests and wetlands. Based on the determined correlation between the two ecological resources, a spatial synergistic planning strategy was proposed. This strategy was combined with grey infrastructure to ensure further enhancement of the synergistic benefits that the two resources exhibit, while reducing their trade-offs. Accordingly, this strategy can be used as NbS for sustainable city development in Tianjin.

The subsequent sections of this paper are organised as follows. Firstly, the research approach of "forest–wetland" NbS is proposed. Secondly, the spatial patterns of forests and wetlands, correlation between the spatial distribution of forests and wetlands, and spatial correlation between the areas of forest growth and wetland growth within a certain distance in different years are studied, which reflects the synergistic and potential trade-offs between forests and wetlands. Finally, the "forest–wetland" spatial planning strategy is formulated based on the core issues of NbS and the spatial analysis result. This paper aimed to construct a research approach of "forest–wetland" NbS. The research will help decision makers develop "forest–wetland" strategies from the perspective of NbS, which is important for sustainable city development.

## 2. Materials and Methods

### 2.1. Study Area

Tianjin (116°43′–118°04′ E, 38°34′–40°15′ N) is in North China. It is one of the core cities of the Beijing–Tianjin–Hebei urban agglomeration, covering an area of 11,966.45 km$^2$ (Figure 2). Tianjin has a warm temperate humid monsoon climate, with an annual mean temperature of 14 °C and a low annual total precipitation of 360–970 mm. Tianjin's rainfall occurs mostly in the summer, and the city is also prone to flooding. The terrain mainly consists of plains and depressions, with low hills to the north. Overall, the elevation gradually decreases from north to south, with the highest elevation in the north, at an altitude of 1052 m. The lowest elevation is in the southeast, at an altitude of 3.5 m. Forest-linked resources in Tianjin are scarce, but an increase in the forest cover has been apparent. The first national forest inventory (1949) revealed that the forest cover of Tianjin was 2.1%; however, it has increased to 12.01%, as of 2018. In terms of water resources, Tianjin is rich in water systems, and these systems include viable wetland resources. The Da Qing River, South Canal, North Canal, Ziya River, and Yongding River converge in Tianjin. However, as a consequence of rapid urban development, the wetland coverage experienced a steep decrease. At the beginning of the 20th century, the wetland coverage in Tianjin was approximately 45.9%, and it dropped to 8.5% in the 1970s [19]. With ecosystem protection now being prioritised, wetlands have been gradually restored. The Second National Wetland Inventory (2013) showed that the wetland coverage reached 24.70%. However, this number is still far from the initial coverage area. Forests and wetlands are

considered NbS for sustainable city development, but overall, these ecological resources exhibit different spatiotemporal dynamics. Therefore, a study on the changes of synergistic patterns of forests and wetlands, using Tianjin as a study site, can provide a scientific basis for the strategic formulation of two variables NbS.

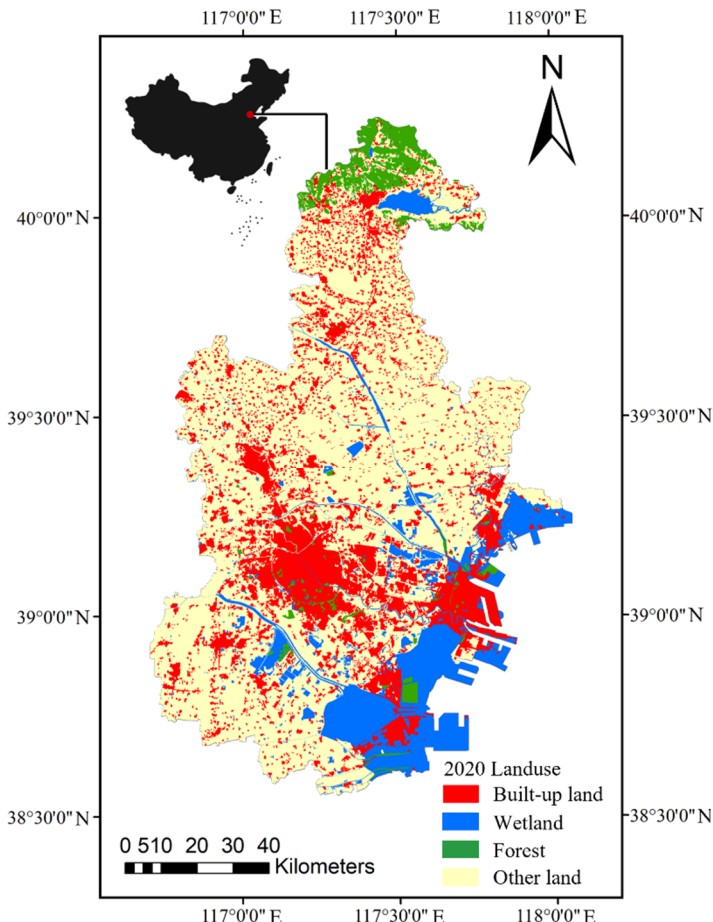

**Figure 2.** Location of Tianjin, China.

### 2.2. Research Approach

NbS has 7 processes, of which the first 3 are identification, analysis, and solution of problems, which are generally consistent with the basic steps of spatial planning. In this study, the main framework of the research approach of "forest–wetland" NbS was established based on the three steps. In addition, NbS has 4 core issues that guide the corresponding solutions which are closely related to the connotation of NbS. The 4 core issues are reflected in the above 3 steps. The research approach of "forest–wetland" NbS entails the following: (1) NbS was proposed to tackle problems of sustainable city development; (2) both forestation and wetland restoration were regarded as NbS for sustainable city development. Forests and wetlands were selected as NbS variables to identify the spatial correlation between forests and wetlands at a macro level. In around 2000, China's ecosystem was severely damaged. Subsequently, in 2008, China proposed the construction of an ecological civilization. By 2020, the ecosystem improved. Multi-temporal studies can help to better perceive the spatio-temporal variation patterns of objects [20]. Therefore, 2000, 2010 and 2020 were selected as sample years to identify the synergistic relationship between forest and wetland growth across various year intervals at a micro level. Additionally, by proposing and exploring potential trade-offs between forests and wetlands; (3) based on the analysis results, the NbS implementation processes were designed, which involved general principles and guidelines for the overall planning strategy, as well as

specific planning strategies. The research approach, which can be applied to any city, is illustrated in Figure 3.

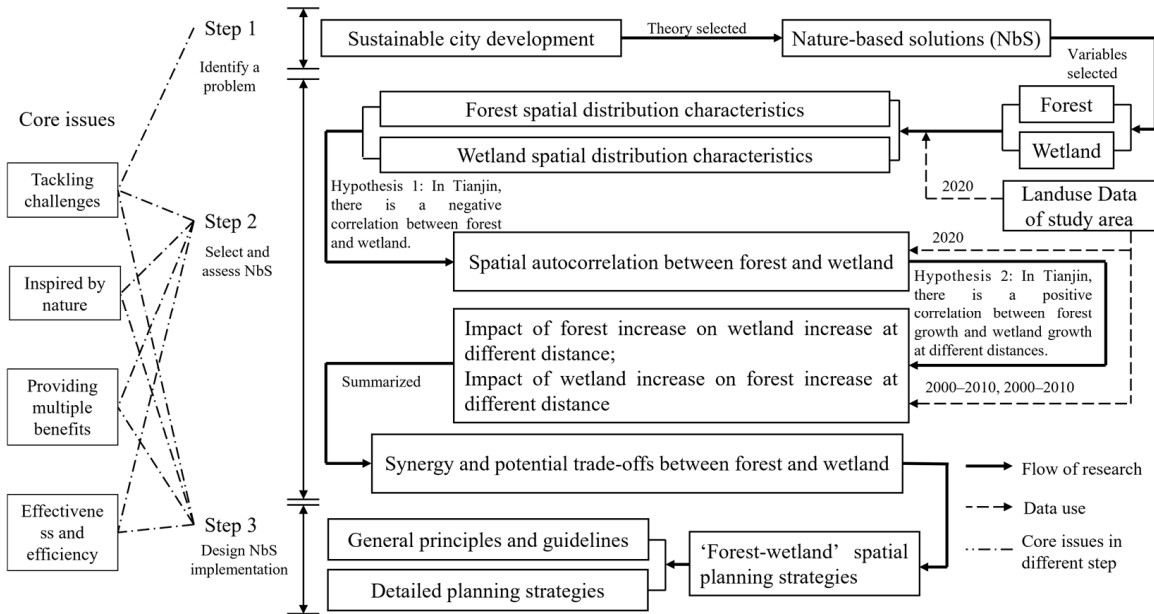

**Figure 3.** Flow chart of research approach of "forest-wetland" NbS.

### 2.3. Data Sources and Database

The total number of water resources and precipitation amounts was obtained from the National Bureau of Statistics of the People's Republic of China (http://www.stats.gov.cn/, accessed on 2 April 2022) and the Bureau of Water Resources of Tianjin (http://swj.tj.gov.cn/, accessed on 2 April 2022). National forest and wetland inventory data were obtained from the Forest Knowledge Service System (http://forest.ckcest.cn/, accessed on 2 April 2022). Land use data for the study area were obtained from GLOBELAND 30 (http://www.globallandcover.com/, accessed on 5 April 2022). Digital elevation model (DEM) data with a resolution of 30 m were obtained from the geospatial data cloud (http://www.gscloud.cn/, accessed on 5 April 2022). DEM and Landsat data were obtained with ArcGIS 10.2 for Desktop software (Environmental Systems Research Institute, Inc. Redlands, CA, USA) and GeoDa 1.18.0 software (Luc Anselin. Chicago, IL, USA).

The statistical data used in this paper were officially provided by the Chinese government, and the geographic information data used in this paper was obtained from the National Aeronautics and Space Administration (NASA) and National Geomatics Center of China (NGCC). In this paper, statistical data and geographic information data are official data; therefore, the accuracy of statistical data was verified by the official sector. The statistical robustness of the database can therefore be guaranteed.

To determine the various parameters such as spatial distribution, concentration degree, kernel density, and spatial autocorrelation, a spatial database of forest/wetland for Tianjin was constructed using the data from 2020. Firstly, a 1 × 1 km grid of Tianjin was created using the fishnet tool via ArcGIS (Data Management Tools—Feature Classes—Created Fishnet). The total number of valid units within the Tianjin area is 12,522. The forest/wetland spatial data were then overlaid with the grid (Analysis Tools-Overlay-Intersect) (Figure 4a); at the same time, the polygon features were converted to point features for the above spatial analysis (Data Management Tools—Features—Feature To Point) (Figure 4b), and the initial and growth areas of the forest/wetland within each unit for the spatial autocorrelation analysis were taken into account (Figure 4c).

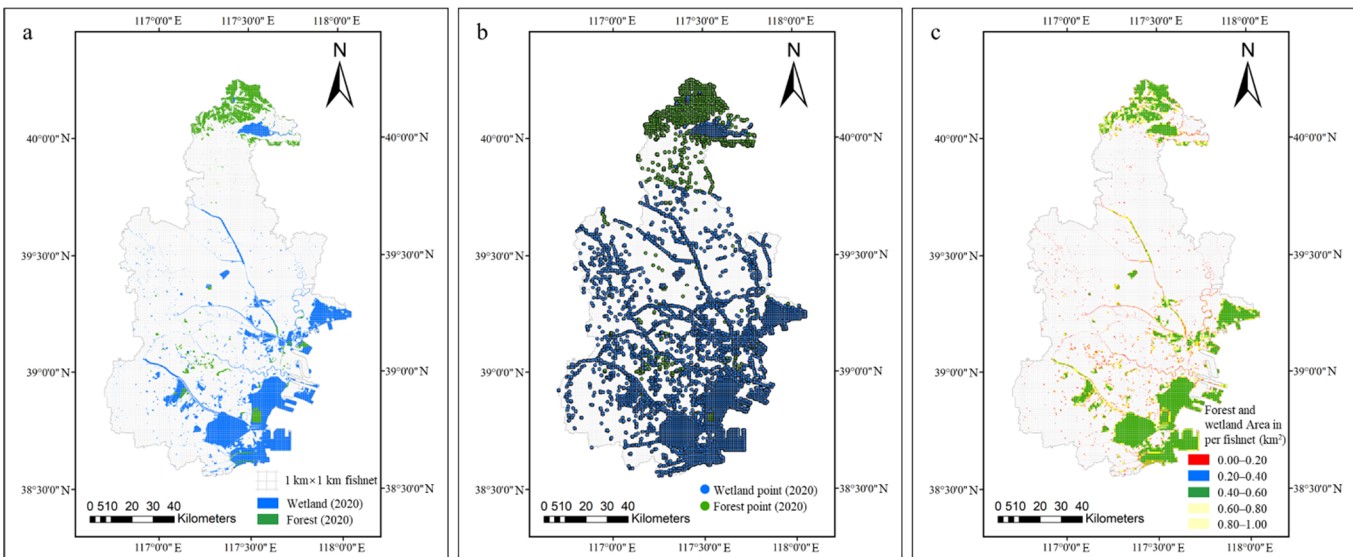

**Figure 4.** The forest/wetland spatial data in Tianjin (2020) (**a**), point features of forest/wetland in Tianjin (2020) (**b**), forest and wetland area in per fishnet (**c**).

## 2.4. Methods

This paper aimed to explore the spatial synergy and potential trade-offs between forests and wetlands. Specifically, the respective spatial patterns of forests and wetlands were identified, followed by a spatial correlation analysis between them. It is advisable to use analysis methods such as spatial distribution pattern analysis, geographic concentration analysis, kernel density estimation, and spatial autocorrelation analysis. Lower total water resources could increase the potential trade-offs between forests and wetlands. Green infrastructure, such as a rainwater storage facility, could increase total water resources to reduce the potential trade-offs. A hydrological analysis was used in this paper to help locate the rainwater storage facility.

### 2.4.1. Spatial Distribution Pattern

Point pattern analyses often focus on spatial distribution patterns [21]. Among them, the nearest neighbour analysis (NNA) is used to determine whether the spatial distribution of point features is clustered, random, or dispersed [22]. In addition, the nearest neighbour index (NNI) is typically used to determine the spatial distribution of point features [23]. The formula for the NNI is as follows:

$$R = \frac{\overline{r}_o}{\overline{r}_e}; \quad \overline{r}_o = \frac{\sum_{i=1}^{n} d_i}{n}; \quad \overline{r}_e = \frac{0.5}{\sqrt{n/A}}$$

where $R$ is the NNI, $\overline{r}_o$ is the average distance between the forest/wetland point features and the nearest point features, $\overline{r}_e$ is the expected average distance between the forest/wetland point features and their nearest point features under ideal conditions, $n$ is the total number of point features and $A$ is the area of Tianjin. If $R = 1$ or close to 1, the spatial distribution pattern of the forest/wetland of Tianjin is random; if $R < 1$, the spatial distribution pattern is dispersed; if $R = 0$, the spatial distribution pattern is completely concentrated. In this study, NNI was calculated using ArcGIS 10.2.

2.4.2. Geographic Concentration

1.  Equilibrium determination

The Geographic concentration index (GCI) was used. The GCI is a common index adopted in point pattern analysis to discern whether point features are spatially distributed in a balanced pattern [24]. The formula for the NNI is as follows:

$$G = \sqrt{\sum_{i=1}^{m} (x_i/n)^2} \times 100$$

where $G$ is the geographic concentration index for Tianjin. The larger the $G$ value, the greater the degree of aggregation of Tianjin's forest/wetland spatial distribution. $x_i$ represents the number of forest/wetland point features in the $i$th unit. $n$ is the total number of forest/wetland point features. $m$ is the total number of units. In this study, NNI was calculated using Microsoft Office 365.

2.  Equilibrium Degree Densification

The Gini index (GI) is a common indicator used to measure the income gap between residents of a country or region and is now applied to analyse spatial distribution differences [25]. The GCI can only reflect whether the distribution of forests/wetlands is in equilibrium. The GI can further measure the equilibrium level of the spatial distribution of forests/wetlands in Tianjin. The formula for the GI is as follows:

$$H = -\sum_{i=1}^{N} P_i ln P_i; \ H_m = ln N$$

$$Gini = H/H_m; C = 1 - Gini$$

where $P_i$ is the proportion of forest/wetland point features to the total number of forest/wetland point features in the $i$th unit. $N$ is the total number of units. $C$ is the equilibrium degree of the spatial distribution of forests and wetlands in Tianjin. $Gini$ is between 0 and 1. A larger value of $Gini$ indicates a higher concentration of forest/wetland spatial distribution in Tianjin. $Gini < 0.2$ is usually considered as absolute equilibrium. If $Gini$ is close to 0, it means that the spatial distribution of the forest/wetland of Tianjin tends to have absolute equilibrium. If $0.2 < Gini < 0.3$, then the spatial distribution of forest/wetland of Tianjin is relatively balanced. If $0.3 < Gini < 0.4$, then the spatial distribution is generally balanced. If $0.4 < Gini < 0.5$, then the spatial distribution is uneven. If $Gini > 0.5$, then the spatial distribution is highly uneven. In this study, $Gini$ was calculated using Microsoft Office 365.

2.4.3. Kernel Density

Kernel density estimation (KDE) is a nonparametric estimation method widely used in the spatial analysis of point data. Its main function is to estimate the density of point patterns [26], reflecting the impact intensity of point features in the neighbourhood. The formula for the KDE is as follows:

$$fx = \frac{1}{wh} \sum_{i=1}^{w} k\left(\frac{x - X_i}{h}\right) \tag{1}$$

where $f(x)$ is the kernel density. A large value represents a dense distribution of forest/wetland points in Tianjin. $k\left(\frac{x-X_i}{h}\right)$ is the kernel function, $h$ is the width of the moving cell, and $h > 0$; $w$ is the quantity of forest/wetland points in a cell; and $x - X_i$ represents the distance from $x$ to $X_i$. In this study, KDE was calculated using ArcGIS 10.2.

### 2.4.4. Spatial Autocorrelation

Spatial autocorrelation analyses focus on the location and specific attributes of a study object. This method can help to accurately describe the spatial distribution pattern of and explain the spatial interactions between elements [27]. Moran's I is a common index of spatial autocorrelation that captures the degree of interdependence between variables. Specifically, Global Moran's I reflects whether spatial clustering or anomalies exist, and Local Moran's I reveals the location where clustering or anomalies occur. In this study, bivariate Moran's I [28] with GeoDa 1.18.0 was used to analyse the spatial correlation between forests and wetlands, which will assist in determining spatial planning and design strategies. The formula for Bivariate Global Moran's I is as follows:

$$I' = \frac{N \sum_{i=1}^{N} \sum_{j=1}^{N} w_{ij}(x_i - \overline{y})(x_j - \overline{y})}{\left(\sum_{i=1}^{N} \sum_{j=1}^{N} w_{ij}\right) \sum_{i=1}^{N} (x_i - x_j)^2}$$

The formula for Bivariate Local Moran's I is as follows:

$$I_i = \frac{x_i - \overline{y}}{\sigma^2} \sum_{j=1}^{N} w_{ij}(x_j - \overline{y})$$

where $I'$ is the bivariate spatial autocorrelation coefficient. $x_i$ and $x_j$ are observations of the same attributes of variables $i$ and $j$ at their own locations, respectively. The forests and wetlands in this study were taken as bivariate variables, and their area was the attribute used for the analysis. $N$ is the sample size. $\overline{y}$ is the average of the second attribute values. $w_{ij}$ is the spatial weight matrix, which is the Euclidean distance used in this study. $I_i$ is the local spatial autocorrelation coefficient of object $i$ at its location in the study area. $\sigma^2$ is the variance of $x$. Consistent with the global Moran's I index, the bivariate Moran's I index has a range of values $[-1, 1]$. If $I < 0$, then there is a negative spatial correlation between the forest/wetland and its neighbouring wetlands/forests; if $I > 0$, then there is a positive spatial correlation between the forest/wetland and its neighbouring wetlands/forests; and if $I = 0$, then the forest/wetland is not spatially correlated with its neighbouring wetlands/forests. The correlation patterns of local spatial autocorrelation can be divided into the following four types: high-high, low-high, low-low, and high-low. Specifically, the high-high mode or low-low mode reflects a positive spatial correlation, whereas the low-high mode or high-low mode denotes a spatial anomaly, which is a manifestation of a negative spatial correlation.

### 2.4.5. Hydrological Analysis

Hydrological analysis is based on a digital elevation model (DEM) data raster to establish a water system model [29], which is used to study the hydrological characteristics and simulation of surface hydrological process, and create a forecast for the surface hydrological situation in the future. The hydrological analysis model can help to analyze the scope of the flood, position the runoff pollution sources, and calculate geomorphological change on runoff. Hydrological analysis includes filling sinks, calculating flow direction, calculating length, calculating flow accumulation, watershed division, river grading, connecting water systems and vectorization. In this paper, flow direction and flow accumulation were calculated by using the hydrologic modelling tools in the ArcGIS spatial analyst extension. The process of Hydrological analysis is as follows: (1) Load DEM and create a raster spatial layer named "Filled DEM"; (2) calculate flow direction and flow accumulation based on "Filled DEM"; (3) export hydrological analysis results.

## 3. Results

### 3.1. Spatial Distribution of Forests

Forest distribution patterns are shown in Figure 5a. The average distance between a forest point feature and its closest neighbouring point feature is 694.0215 m, and the

expected average distance is 1441.3565 m. R = 0.481506, which is the value of NNA < 1. The results showed that the spatial distribution patterns of forests in Tianjin are clustered.

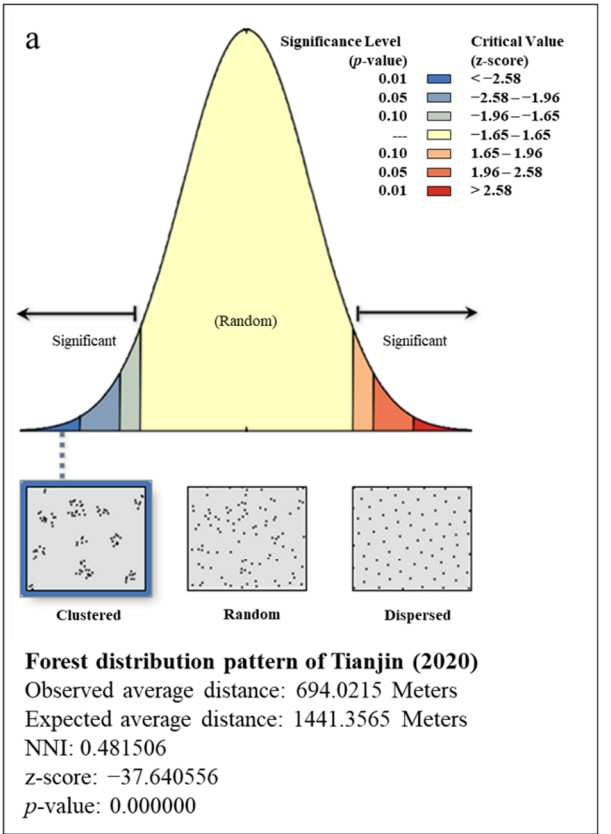 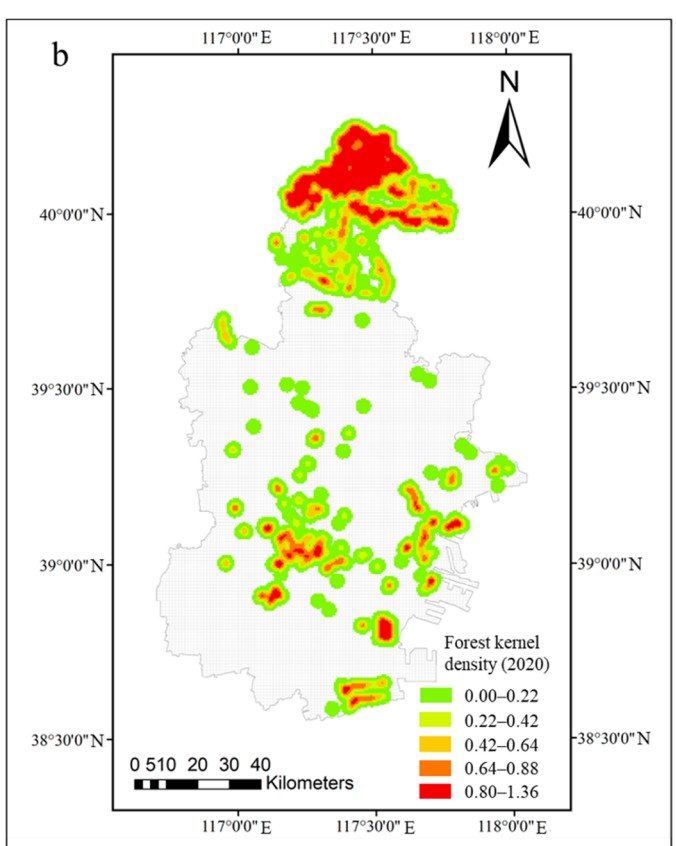

**Figure 5.** Forest distribution pattern of Tianjin (2020) (**a**), forest kernel density of Tianjin (2020) (**b**).

The GCI was calculated as G = 2.6371. If all forest point features of Tianjin were evenly distributed in the grid, its GCI, $\overline{G}$ = 0.8936. $G > \overline{G}$ indicated that the Tianjin forests appeared to show a concentrated and unevenly distributed spatial pattern. The GI of forests in Tianjin was also measured. *Gini* = 0.8851 and *Gini* > 0.5 indicate that Tianjin has an uneven spatial distribution of forests.

The Kernel Density tool of ArcGIS (Spatial Analyst Tools—Density—Kernel Density) was used to calculate the kernel density of the forest space of Tianjin (Output cell size = 500 m, Search radius = 2500 m). The kernel density of the forest was classified into the following five categories: extremely high, high, medium, low, and extremely low based on natural breaks (Figure 5b). The distribution of forest resources varies greatly from north to south, with a point pattern in the south and a net pattern in the north. Forest resources are mainly located in the low-mountain hilly areas of the Jizhou administrative district.

### 3.2. Spatial Distribution of Wetlands

Wetland distribution patterns are shown in Figure 6a. The average distance between a wetland point feature and its closest neighbouring point feature is 55.6119 m, and the expected average distance is 821.3076 m. R = 0.798254, which is the value of NNA < 1. The results showed that the spatial distribution patterns of wetlands in Tianjin were clustered.

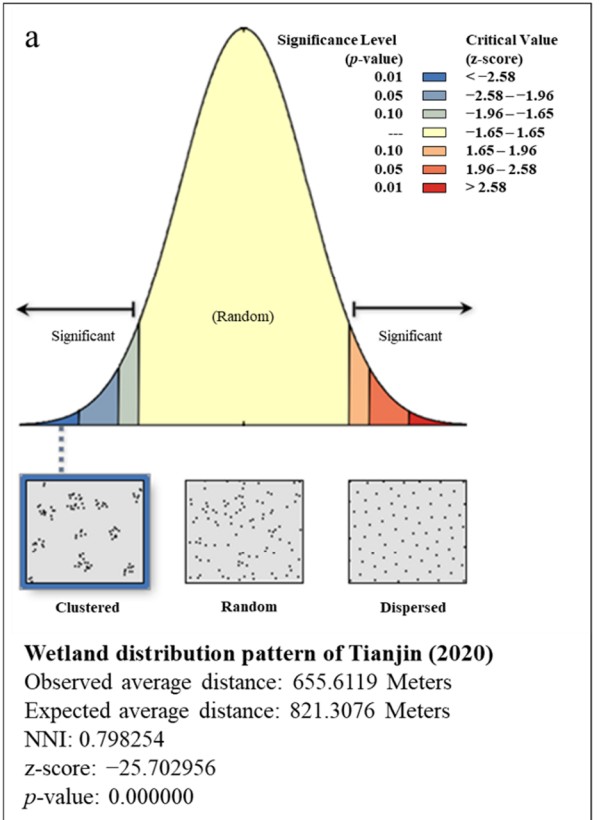

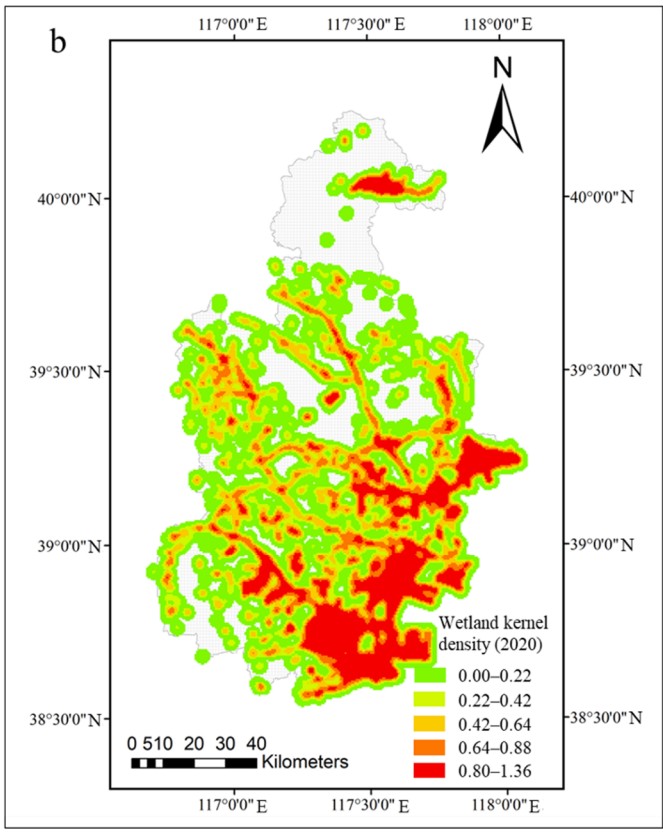

**Figure 6.** Wetland distribution pattern of Tianjin (2020) (**a**), wetland kernel density of Tianjin (2020) (**b**).

The GCI was calculated as G = 1.5017. If all the wetland point features of Tianjin were evenly distributed in the grid, its GCI, $\overline{G}$ = 0.7362. $G > \overline{G}$ indicated that the wetlands of Tianjin appeared to be concentrated and unevenly distributed. The GI of wetlands in Tianjin was measured. *Gini* = 0.5695 and *Gini* > 0.5, indicating that Tianjin has an uneven spatial distribution of wetlands. Compared with forests, the spatial distribution of wetlands in Tianjin was more balanced.

The kernel density tool of ArcGIS (Spatial Analyst Tools—Density—Kernel Density) was used to calculate the kernel density of the wetland space of Tianjin (Output cell size = 500 m, Search radius = 2500 m). The kernel density of the wetland was classified into five categories, namely, high, high, medium, low, and low based on natural breaks (Figure 6b). The distribution of wetland resources varies greatly from north to south, with a net pattern concentrated in the south and a point pattern concentrated in the north. The wetland resources are mainly located in the Binhai New Area, Jinghai, Xiqing, Baodi, Jizhou, and other administrative districts. This is because these areas are rich with water systems that provide a sound ecological basis for wetland conservation.

### 3.3. Macro Spatial Autocorrelation between Forests and Wetlands of Tianjin

#### 3.3.1. Bivariate Global Moran's I

Based on the computation of the spatial distribution of forests and wetlands, we found that their distribution patterns showed roughly opposing trends. Therefore, assumption I was proposed, that there is a negative correlation between the spatial distribution of forests and wetlands in Tianjin. To test assumption I, spatial correlation between forests and wetlands in 2020 was analysed using area as a variable. Forest and wetland databases were processed using Geoda 1.18.0, and the results showed that each forest/wetland patch had at least one adjacent patch with a minimum Euclidean distance of 9949.08 m. Therefore, the specified bandwidth could be set to 10,000 m. Using the Bivariate Global Moran's I index analysis tool (Spatial Analysis—Bivariate Global Moran's I), the forest and wetland

areas within each unit were calculated as the first and second variables, respectively. The scatter chart (Figure 7a,b) of the Bivariate Global Moran's I between forest-wetland and wetland-forest areas is shown below. The value of the Bivariate Global Moran's I between forest-wetland areas was −0.156, and the value of the Bivariate Global Moran's I between wetland-forest areas was −0.153. This indicated that the forest–wetland areas show a negative spatial correlation within Tianjin, thereby supporting assumption I. Accordingly, it can be inferred that forests and wetlands cannot form an effective blue-green ecological network in Tianjin.

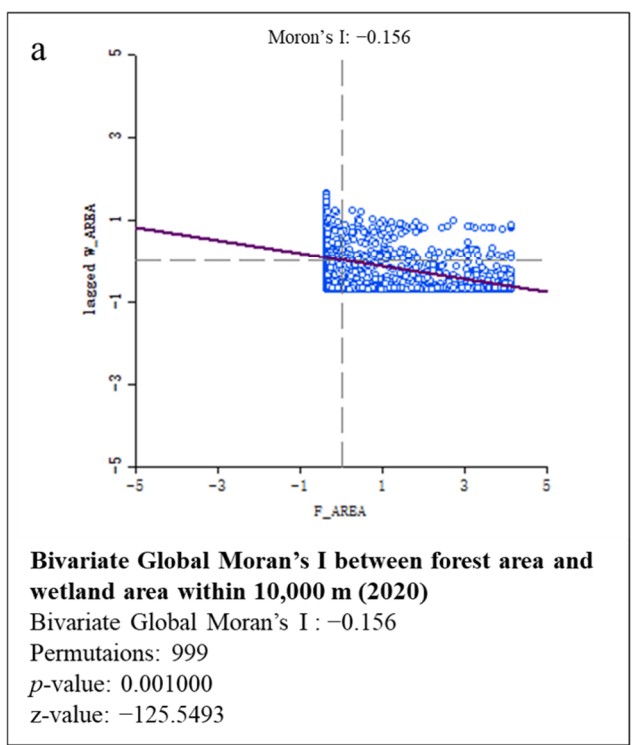

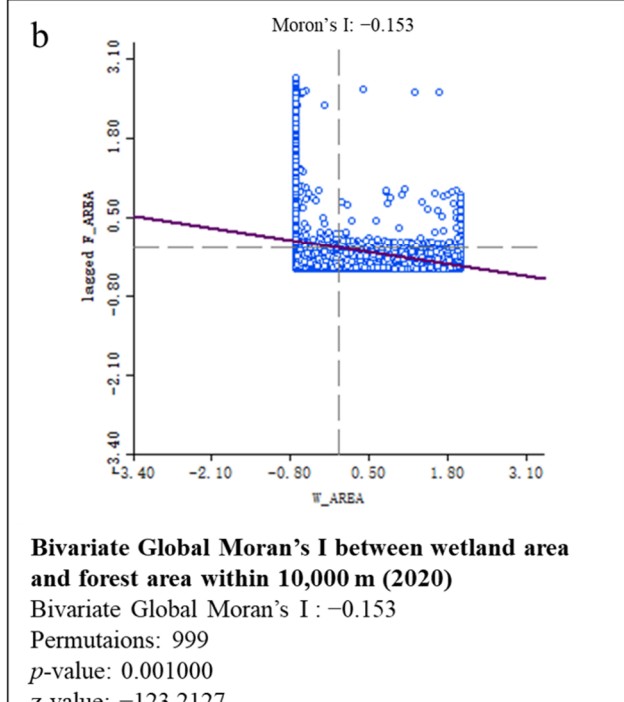

**Figure 7.** Bivariate Global Moran's I between forest area and wetland area within 10,000 m in Tianjin (2020) (**a**), Bivariate Global Moran's I between wetland area and forest area within 10,000 m in Tianjin (2020) (**b**).

### 3.3.2. Bivariate Local Moran's I

Further analysis of the spatial divergence of agglomeration was performed using a bivariate local Moran's I. The forest and wetland areas within each unit were calculated as the first and second variables, respectively. The Bivariate Local Moran's I Index Analysis Tool of Geoda (version 1.18.0) software (Spatial Analysis—Bivariate Local Moran's I) was used to obtain the Lisa cluster map of the forest–wetland areas (Figure 8a) and wetland–forest areas (Figure 8b). High–high areas were identified as hotspots. The LISA clustering map illustrates that there are only a few high forest area and high wetland area areas in Tianjin, and this type of area is mainly concentrated in the Yuqiao Reservoir, Tuanbo Reservoir, and Binhai New Area (Figure 8a). The few high wetland area and high forest area areas are mainly located near Panshan Mountain and the Binhai New Area (Figure 8b). The high–low area indicates that forest/wetland has a large area, whereas the wetland/forest has a small area within a 10,000 m range. Compared with low–low areas, a high–low area can be transformed into hotspots more efficiently and have the potential to improve the quality of Tianjin's blue-green ecological network. Therefore, priority should be given to the high–low areas when preparing the "forest–wetland" spatial planning strategy.

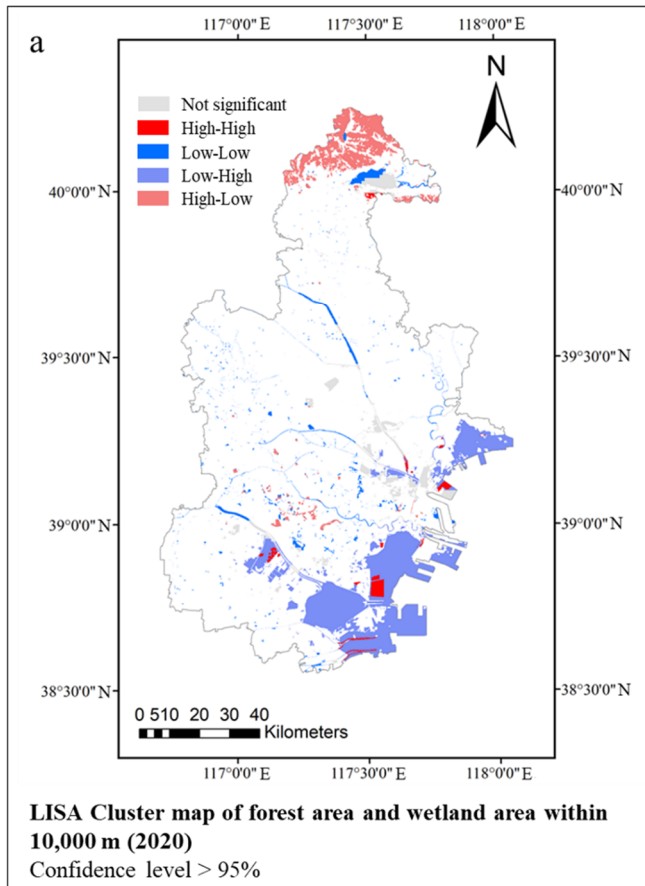
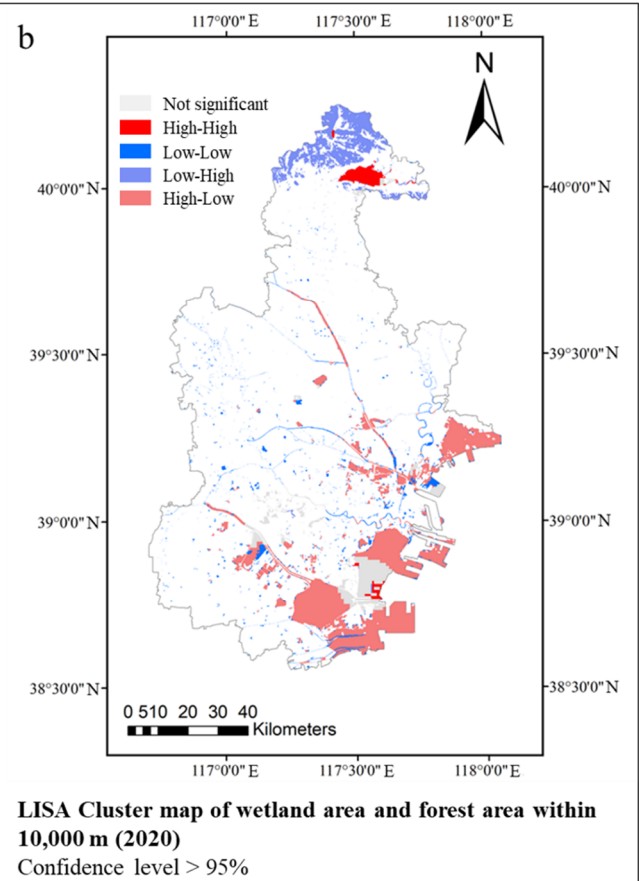

**Figure 8.** LISA Cluster map of forest area and wetland area within 10,000 m in Tianjin (2020) (**a**), LISA Cluster map of wetland area and forest area within 10,000 m in Tianjin (2020) (**b**).

*3.4. Micro Spatial Autocorrelation between Forest and Wetland at Different Distance*

A previous analysis revealed that the forests of Tianjin are mainly concentrated in the northern part of the city, and that the wetlands are mainly concentrated around the water system in the south. Spatial distribution of forests and wetlands revealed that forests and wetland areas were negatively correlated throughout Tianjin. However, because only forests and wetlands of Tianjin were analysed in 2020, it was still not clear whether there is a synergy or trade-off between forest and wetland growths over time. To further validate the relationship between spatial forest and wetland growths, we proposed the following assumption: there is a positive correlation between forest/wetland and wetland/forest growths within is a certain spatial distance threshold, and vice versa, showing a negative correlation. Therefore, we further analysed the interrelationships between forest and wetland growth areas at different distances in 2000–2010 and 2010–2020.

3.4.1. Impact of Forest Increase on Wetland Increase at Different Distances

Geoda software (1.18.0) was used for a Bivariate Global Moran's I index analysis (Spatial Analysis—Bivariate Global Moran's I). The forest and wetland growth areas within their respective unit were calculated as first and second variables. The impact of forests on wetlands increased at different distances, as illustrated in Figure 9. Figure 9a illustrates that the growth of forests was positively correlated with the growth of wetlands within a distance of 0–400 m from 2000 to 2010. This indicates that as the forest area increased, the area of wetlands surrounding the forests also increased. When the distance exceeded 500 m, the growth of the forest area was negatively correlated with that of the wetland area. This indicates that as the forest area increased, the wetland around the forest showed a decrease. Figure 9b shows that the growth of forest area was positively correlated with

the growth of wetland area, within a distance of 0–600 m, from 2010 to 2020. When the distance exceeded 700 m, the growth of the forest area was negatively correlated with that of the wetland area.

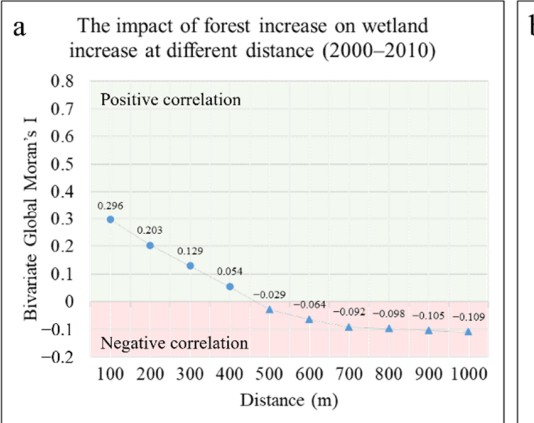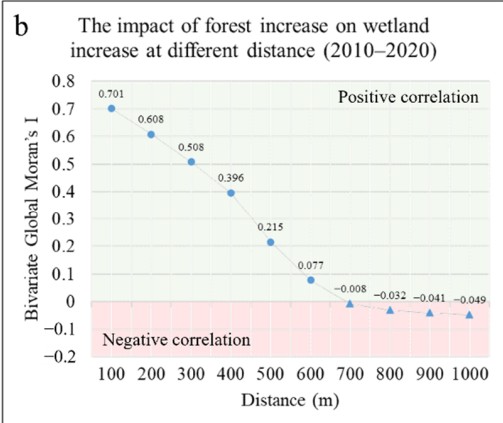

**Figure 9.** Bivariate Global Moran's I between forest increase and wetland increase at different distance in Tianjin, from 2000–2010 (**a**), Bivariate Global Moran's I between forest increase and wetland increase at different distance in Tianjin, from 2010–2020 (**b**).

3.4.2. Impact of Wetland Increase on Forest Increase at Different Distance

Using the same analysis tool, the wetland growth area within each unit was calculated as the first variable, and the forest growth area was calculated as the second variable. The impact of wetlands on forests increased at different distances, as illustrated in Figure 10. Figure 10a illustrates that the growth of wetlands was positively correlated with the growth of forests, within a distance of 0–400 m from 2000 to 2010. This indicates that, as the wetland area increased, the area of forests surrounding the wetland also increased. When the distance exceeded 500 m, the growth of the forest area was negatively correlated with that of the wetland area. This indicates that as the area of forests increased, the wetlands around the forest showed a decrease in area. Figure 10b shows that the growth of wetland areas was positively correlated with the growth of forest areas, within a distance of 0–600 m from 2010 to 2020. When the distance exceeded 700 m, the correlation was negative.

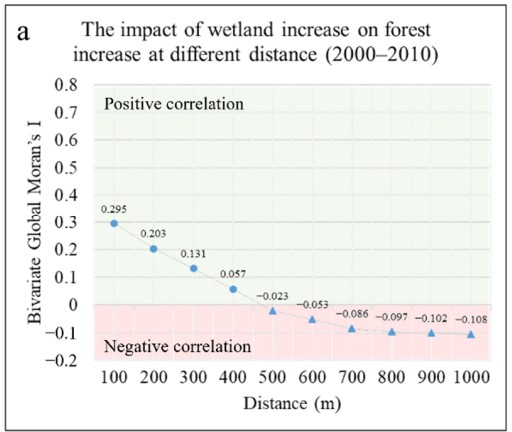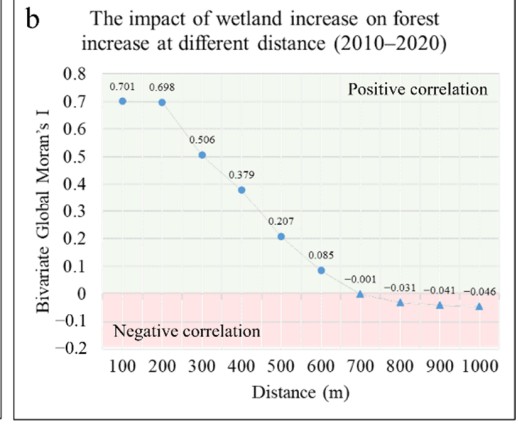

**Figure 10.** Bivariate Global Moran's I between wetland increase and forest increase at different distance in Tianjin, from 2000–2010 (**a**), Bivariate Global Moran's I between wetland increase and forest increase at different distance in Tianjin, from 2010–2020 (**b**).

3.4.3. Synergy and Potential Trade-Off between Forest and Wetland at Different Distance

Figures 9 and 10 imply that as the area of forests and wetlands increase, the quality of forest and wetland ecosystems improves. This means that the synergy between the

growth of forest areas and growth of wetland areas gradually increases, and the trade-off relationship gradually decreases. This result supports assumption II. Therefore, in the "forest–wetland" spatial planning strategy, wetlands/forests can be preferentially located around forests/wetlands to capitalize on their synergistic effect. Ideally, the optimal distance between the forest and wetland should be kept within 400 m. In cases where a barrier is created by built-up areas or agricultural land, the distance between forests and wetlands should not exceed 700 m.

In the Beijing–Tianjin–Hebei urban agglomeration, forest growth has not yet resulted in wetland losses [15]. However, strategies should be developed to prevent trade-offs between forests and wetlands. The amount of water available for ecological resources in Tianjin is insufficient [30]. An increase in forest area will lead to an increase in evaporation, which will translate into a greater water demand. This will reduce the total water entering the wetlands, which in turn will hinder the growth of wetlands in Tianjin. NbS supports the synergy of "green-blue-grey" infrastructure to achieve better NbS performance [6]. To reduce potential trade-offs between forests and wetlands, it is necessary to use grey infrastructure, such as water tanks, to collect rainwater and increase the total water resources, to ensure that sufficient water is channeled towards the ecological resources of Tianjin.

### 3.5. Hydrological Analysis of Tianjin

Using the digital elevation model (DEM), the rainwater flow direction of Tianjin (Figure 11a) was analysed via ArcGIS (Spatial Analyst Tools—Hydrology). This was performed to determine the flow direction of rainwater in each region of Tianjin. We found that the high-level rivers of Tianjin tend to have intense flows and also are dense (Figure 11b), providing a good foundation for the aquatic ecosystem of Tianjin. However, this also makes flood-prone areas prevalent in the region. To reduce the risk of flooding under extreme precipitation, the Tianjin Municipal Water System Plan (2008–2020) delineates several flood storage and detention basins as flood buffers (Figure 11c). However, the function of these basins is generally to mitigate the impact of flooding on cities; even though through the lens of NbS, these basins do not offer multiple benefits. These basins may generate better value for sustainable city development if they are synergised with other grey infrastructure.

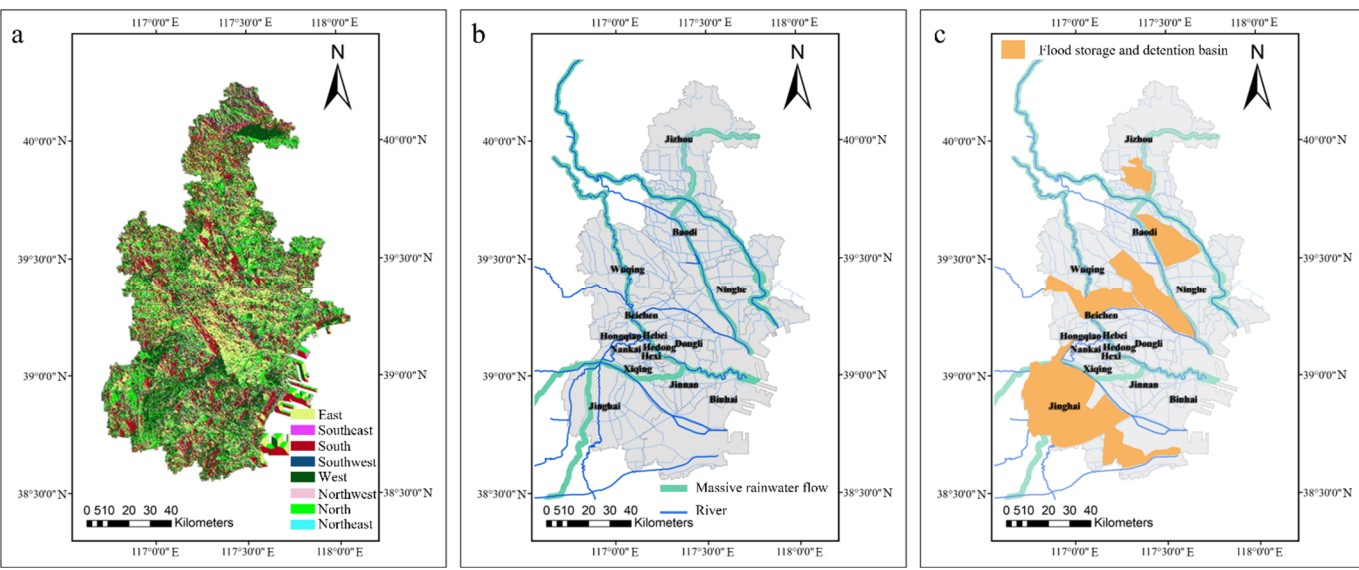

**Figure 11.** Rainwater flow direction of Tianjin (**a**), flow accumulation of Tianjin (**b**), flood storage and detention basins of Tianjin (**c**).

### 4. Discussion

Based on the NbS concept, the spatial distribution characteristics of forests and wetlands in Tianjin, and the spatial correlation between them were analysed. By incorporating

the results, feasible "forest-wetland" spatial planning strategies were explored in accordance with the NbS theory to promote the sustainable development of Tianjin.

*4.1. General Principles and Guidelines for Planning Strategy*

4.1.1. General Principles

The principles of the "forest–wetland" spatial planning strategy were defined based on the four core issues of NbS [6,8].

1.  Tackling challenges: To holistically address challenges tied to sustainable city development and to form a complete forest-wetland ecological network.
2.  Inspired by nature synergy: The strategy should follow the synergistic relationship between forests and wetlands, to ensure that they efficiently promote each other.
3.  Providing multiple benefits: The new forest/wetland is not only meant to serve as an ecological substrate for the city, but as a public feature that provides socioeconomic benefits to the general population.
4.  Effectiveness and efficiency: To avoid potential trade-offs between forests and wetlands, the effect of grey infrastructure should be considered in an integrated manner. The synergy of "Blue-Green-Grey" facilities enhances NbS performance.

4.1.2. General Guidelines

The guidelines for "forest–wetland" spatial planning strategies were proposed to reflect the core issues of NbS in the planning process and to help planners and decision makers to understand how to implement the planning strategies (Figure 12).

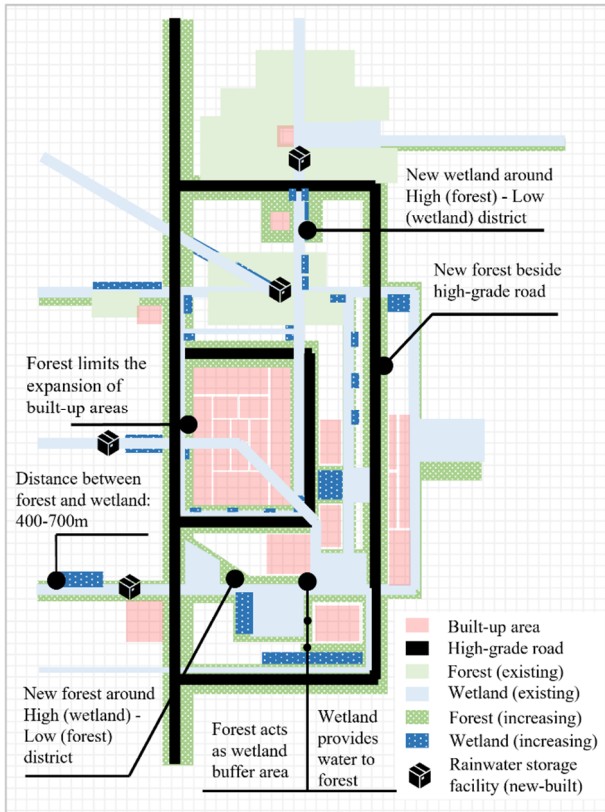

**Figure 12.** Guidelines of "forest-wetland" spatial planning strategies.

1.  An increase in the number and area of forests. To promote the construction of forest cities, including building urban forest parks, forested areas around cities, and protective forest belts; overall, to increase the number of forest patches. These forest patches would act as stepping stones to improve north–south forest connectivity.

Protective forest belts should be preserved in mountains, waterways, and roads to form ecological corridors and serve as a "green-blue ecological corridor" for biological circulation and ecological connectivity [31].

2.  On the one hand, the number and area of wetlands should be increased, and on the other hand, sufficient space and basic conditions for wetland development should be set aside. Increasing the number of wetland patches should be based on current water systems. Combined with forest construction, forests should be used as an ecological buffer for wetlands to guarantee the conservation and restoration of wetland ecosystems, which could protect and enhance water quality [32]. The optimal distance between the forest and wetland should not be more than 400 m.

3.  Promoting the synergistic benefits of forests and wetlands by leveraging their positive spatial correlation. It is recommended that newly created forests be located adjacent to wetlands and vice versa. In some cases, the distance can be adjusted appropriately, but it should not exceed 700 m. Based on this proximity, forests and wetlands can be integrated to build Tianjin's blue–green ecological network.

4.  Based on the hydrological analysis and distribution of flood storage areas, new rainwater harvesting measures were proposed. In combination with the "Forest–Wetland" spatial planning strategy, rainwater should be harvested during periods of abundance to reduce the impacts of urban flooding [33]. Accordingly, rivers should be recharged during periods of water shortage to ensure that the ecological environment uses water for natural restoration.

### 4.2. "Forest-Wetland" Spatial Planning Strategies Inspired by Nature Synergy

#### 4.2.1. Forest-Wetland Ecological Network

The results revealed that forests and wetlands were concentrated in the north and south of Tianjin, respectively. Tianjin lacks wetlands in the north and forests in the south, while the central city of Tianjin lacks both forests and wetlands. Therefore, spatial planning strategies should increase the number of wetlands and forests in accordance with the resource-linked features of each region. As per the macro spatial autocorrelation between forests and wetlands in Tianjin, certain high–low districts have great potential to develop forests and wetlands. Therefore, priority should be given to expanding forests and wetlands around the high–low districts. Meanwhile, the micro-spatial autocorrelation between forest and wetland at different distances demonstrates that the synergistic effect of forests and wetlands mainly exists at a distance of 0–400 m. When ecosystem conditions improve, the distance can be extended from 0 to 700 m. In the spatial planning strategy, new forests/wetlands should be adjacent to wetlands/forests to maximise their synergistic effect (Figure 13). Based on the above factors, a forest–wetland ecological network was established to fill the gap between the current forests and wetlands of Tianjin and to promote synergy between them.

#### 4.2.2. Functions of Recreation, Biodiversity, and Economy

The forest–wetland ecological network promotes synergy between two ecological resources, even though this network performs only a single function (i.e., improving ecosystem quality). However, good NbS requires multiple benefits. Therefore, other factors must be included in the construction of forest wetlands. Taking forestation as an example (Figure 14), the new forests around the central city are mainly positioned as country parks, such as the Xiqing Country and Northern Country Parks. The 10 km service scope covers more than 70% of the central city, which can offer public spots for leisure and entertainment. The forests on the fringes of the central city, such as Jizhou, Jinghai, and Ninghe forests, can be intercropped with vegetation to improve biodiversity and agricultural income [7]. The forest next to the expressway constitutes the green–blue ecological corridor, which mainly plays the role of limiting the sprawl of the urban built-up area. Integrating different types of forests produces more benefits, such as recreation, biodiversity, and economy, in addition to their ecological advantages.

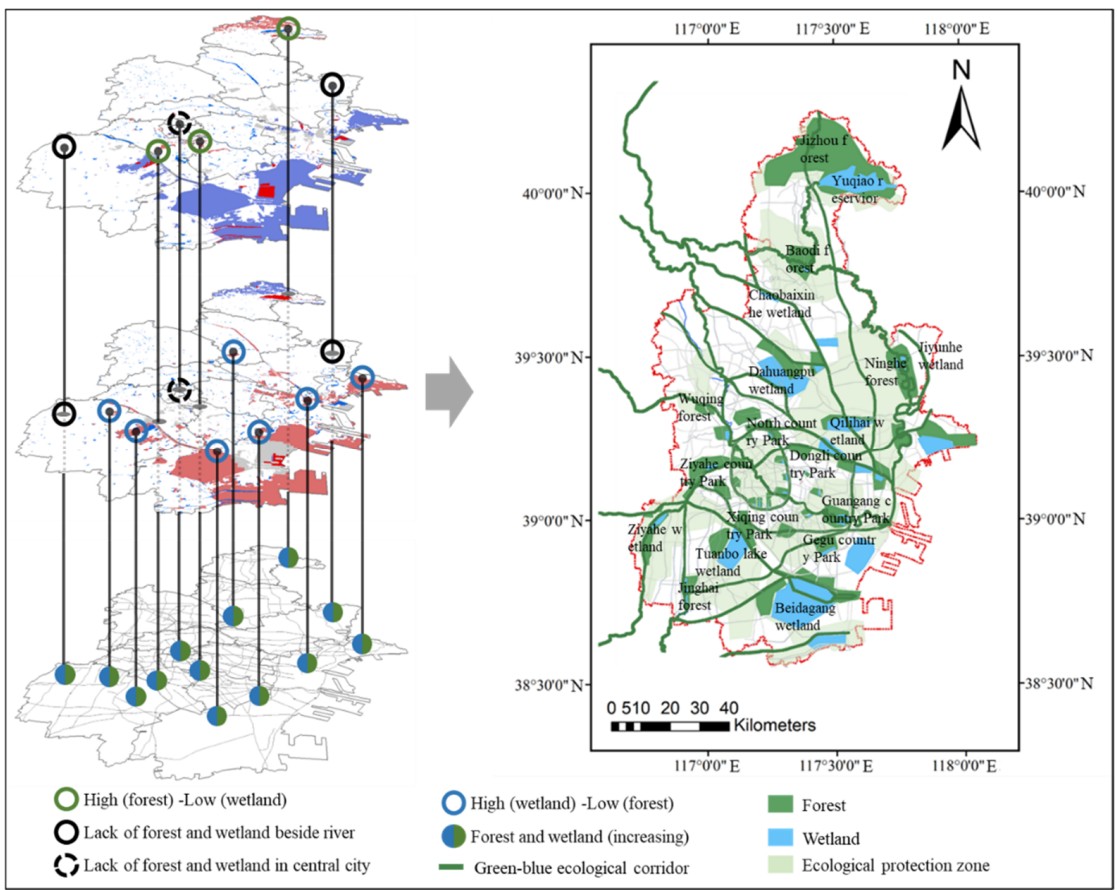

**Figure 13.** Forest-wetland ecological network in Tianjin.

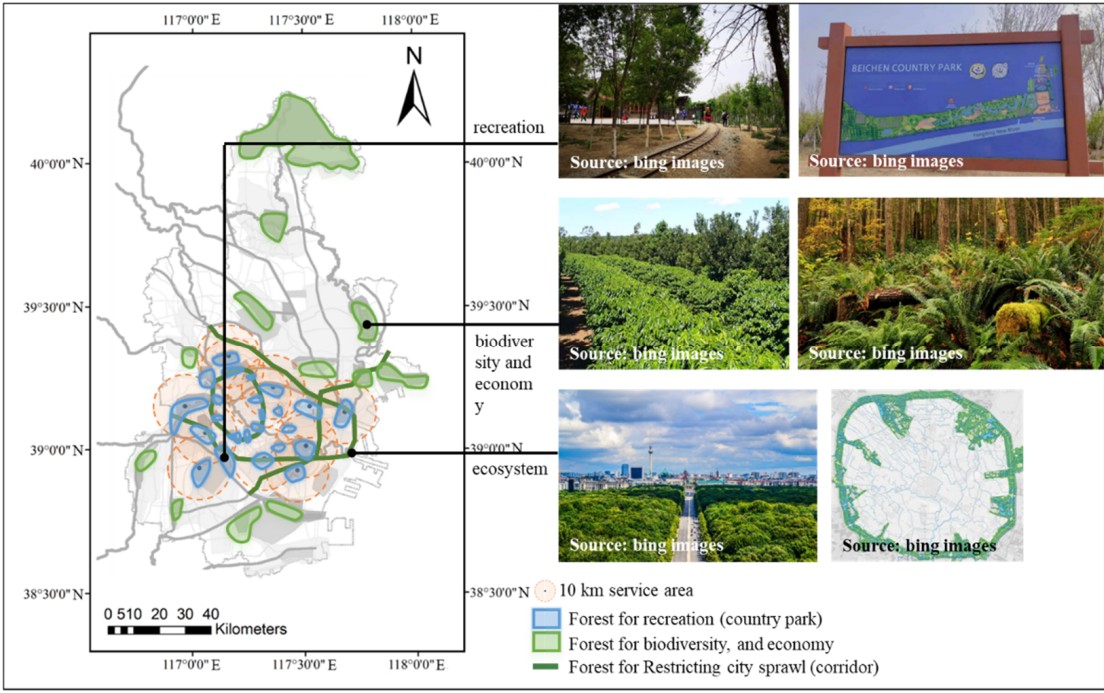

**Figure 14.** Different benefits of different forests.

### 4.3. Grey Infrastructure to Reduce Potential Trade-Off between Forest and Wetland

The increase in both forest [15] and intercropping vegetation area [34] will reduce the total ecological water (ecological environment water use) available to wetlands, which can intensify water-induced trade-offs between forests and wetlands. Therefore, a reasonable total ecological water strategy is necessary to ensure the restoration of Tianjin's forests and wetlands. A study showed that the total ecological water used in Tianjin in 2020 was as high as 1.408 billion $m^3$ [30]. However, the Tianjin Water Resources Bulletin showed that the total water used in the Tianjin ecological environment in 2018 was approximately 557 million $m^3$. These data indicate that the water demand of the ecosystem is far from sufficiently met, and this can severely impede the maintenance of ecosystem services. The water yield coefficient (WYC) refers to the proportion of the total water resources in total precipitation [35]. A low WYC accounts for the low total water used in the ecological environment of Tianjin. Taking Sichuan Province as an example, total water resources (295.264 billion $m^3$) accounted for 57.96% of the total precipitation (509.393 billion $m^3$) in 2018. Assuming that Tianjin can achieve the same WYC (57.96%) by enhancing its rainwater collection capacity, the total water resources of Tianjin will reach 4.019 billion $m^3$ in 2018. Similarly, the ecological water supplement will also increase dramatically and meet the established standards (Table 1).

**Table 1.** Comparison of water resources of Tianjin municipality and Sichuan Province in 2018 (billion $m^3$).

| Province or Municipality | Precipitation | WYC | Total Water Resources | Percentage of Ecological Environment Water to Total Water Resources | Ecological Environment Water |
|---|---|---|---|---|---|
| Sichuan | 5093.93 | 57.96% | 2952.64 | | |
| Tianjin | 69.35 | 25.34% | 17.58 | 31.68% | 5.57 |
| Tianjin * | 69.35 | 57.96% * | 40.19 * | 31.68% | 12.73 * |

*: Assuming that Tianjin has same WYC with Sichuan in 2018.

There are several reasons for the low WYC in Tianjin. On one hand, the percent of impervious surface in Tianjin is high, which reduces the recharge of surface water and groundwater, thus intensifying the evaporation of rainwater, leading to a wastage of rainwater resources. However, the spatiotemporal distribution of precipitation in Tianjin was uneven and mainly concentrated in July and August. Owing to a lack of reasonable collection facilities, rainwater usually causes flooding in Tianjin. Rainwater is evacuated through flooding measures and eventually enters the Bohai Sea, resulting in a waste of rainwater that could have instead been collected. With the new rainwater harvesting facility (Figure 15), rainwater can be collected and stored during the rainy season so that the river can be recharged during droughts. The configuration of rainwater storage facilities should be based on an hydrological analysis, and priority should be given to areas with a high stormwater flow. The rainwater storage facilities are organised into I, II, and III (I > II > III in terms of volume) levels from upstream to downstream according to the number of rivers forming the Tianjin rainwater storage facility system (Figure 16). In addition to forests and wetlands, these facilities can recharge the next level of facilities through rivers.

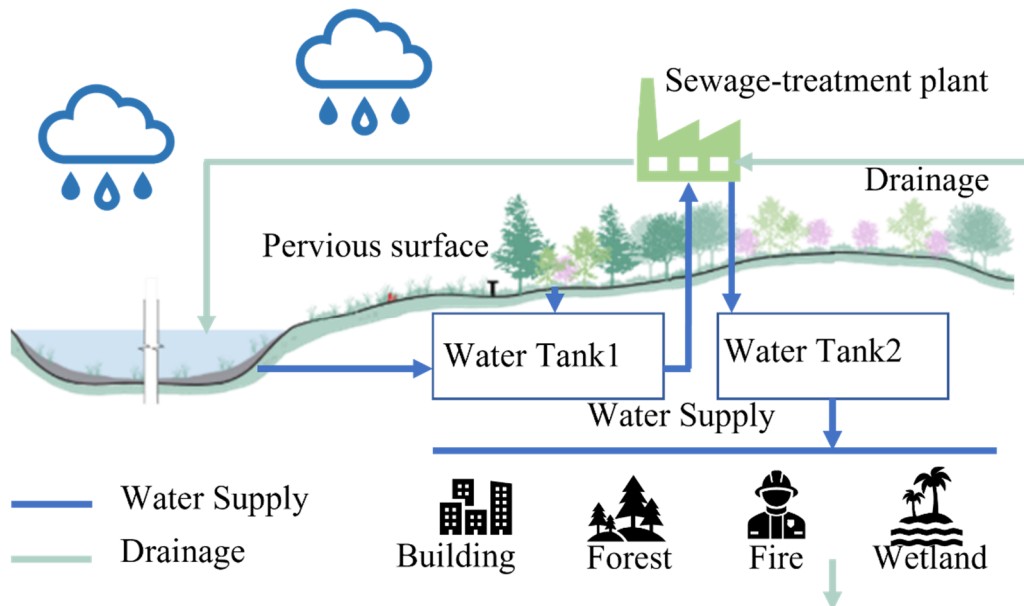

**Figure 15.** Function of rainwater storage facilities.

**Figure 16.** Rainwater storage facility spatial planning in Tianjin.

## 5. Conclusions

In this study, we explored the possible synergy between forests and wetlands in Tianjin for sustainable city development through the lens of the two variables of NbS. (i) Analysis of the spatial distribution of forests and wetlands demonstrated that these ecological resources in Tianjin were concentrated in the north and south of the city, respectively; additionally, they could not form an effective forest-wetland ecological network. (ii) Assumption I was proposed, which states that: "forests and wetlands are negatively correlated in the whole area of Tianjin". The results of the spatial correlation analysis of forests and wetlands in the entire area of Tianjin support this assumption. At the same time, the study proved the lack of a forest–wetland ecological network in Tianjin. (iii) To further explore the relationship between forests and wetlands at smaller scales, assumption II was proposed which states that: "there is a positive correlation between forest/wetland growth and wetland/forest growth over a specific distance range". A spatial correlation analysis revealed that forest and wetland growths were positively correlated in the range of 400 m during 2000–2010. With the restoration of forests and wetlands, the growth of these ecological resources positively correlated in the range of 600 m during 2010–2020. The results supported the assumption that forests and wetlands had a synergistic effect on each other at certain distances. (iv) In northern and northeastern China, the increase in forests has caused a significant decrease in wetlands [15]. The situation has changed in the Beijing–Tianjin–Hebei Urban agglomeration, mainly due to the fact that the Beijing–Tianjin–Hebei urban agglomeration is currently rich in water resources, and its forest area is small, indicating that forest–wetland trade-offs are not currently noticeable. The continued expansion of forests and wetlands is expected to increase the potential for competition between these two ecological resources for water. (v) Based on the four core elements of NbS, the principles of a forest–wetland spatial planning strategy were proposed, and a guiding map of this strategy was constructed. Based on a natural synergy, forest–wetland ecological networks were constructed. With the intent of providing multiple benefits, different types of forest management approaches were proposed to maximise the benefits of forests across multiple spheres, namely, recreation, biodiversity, and economy. Based on a hydrological analysis and forest–wetland spatial planning strategy in Tianjin, three-level rainwater storage facilities I, II, and III were constructed to collect rainwater during periods of heavy precipitation and provide ecological water for forests and wetlands during droughts.

This paper attempted to further NbS research by applying two variables rather than a single variable. By analyzing the synergy and potential trade-offs, a "forest–wetland" planning strategy was proposed to promote the synergy between forests and wetlands, and grey infrastructure was suggested to reduce potential trade-offs between them. Overall, the research can be applied to other cities or regions, and can also be used as a reference for multiple-variables NbS studies.

There are still some deficiencies in this paper, especially in the integration of NbS. Current research on NbS is mostly focused on a single variable and does not consider the correlation between multiple variables. This paper mainly focused on forests and wetlands, and it entailed the exploration of their interrelationships within the context of NbS. Whereas the EC's NbS integrates social, economic, and environmental co-development, NbS was extensively used to address various urban challenges. The use of multiple variables will become a trend in NbS research. For future studies, the inclusion of more variables, such as urban parks, squares, green spaces, river systems, and lakes, will allow for a much more nuanced analysis of the system, which will translate to a better exploration of the synergy and trade-offs among multiple variables, culminating in more robust NbS.

**Author Contributions:** Y.L., conceptualization, methodology, software, formal data analysis, data curation, writing initial draft preparation, and project administration; T.C., supervision and project administration; G.W. and R.Z., validation, and interpretation of results; L.Z. and L.Y., reviewing and editing of the manuscript. All authors have read and agreed to the published version of the manuscript.

**Funding:** This research was funded by the International Cooperation and Exchanges NSFC, Grant agreement ID 52061160366; National Natural Science Foundation of China (NSFC), Grant agreement ID 52078329, China National Key R&D Program during the 13th Five-year Plan Period, Grant agreement ID 2018YFC0704603.

**Institutional Review Board Statement:** Not applicable.

**Informed Consent Statement:** Not applicable.

**Data Availability Statement:** All supporting data are cited in Section 2.

**Conflicts of Interest:** The authors declare no conflict of interest.

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
