# Peer review of "Nature-Based Solutions in “Forest–Wetland” Spatial Planning Strategies to Promote Sustainable City Development in Tianjin, China"

_land, doi:10.3390/land11081227_

Round 1
Reviewer 1 Report
The issue dealt with in the manuscript is current and interesting. The paper has a proper structure. Minor revisions are requested before the final publication.
In the Abstract it is advisable to add a brief description of the methodological approach used in the research and the main results obtained.
In order to improve the Introduction section, it is recommended to add a small paragraph with the explanation of the contents of the subsequent sections.
In section 2 “Materials and Methods” the Figure 3 should be modified. In fact, it is very full of information and not very clear. It is suggested to use the steps of the process (from 1 to 3) as milestones for the graph and to include different forms for different meaning (for example why the years are important? and what is their meaning in the chart?). At least, the verification of the flow of all the arrows should be carried out: it must have a logic sense.
In the subsection 2.3, it is advisable to add some information about the data collected and about any operation has been performed in order to test the statistical robustness of the database.
It is also important to check if all the terms used in the equations are properly introduced and defined.
Section 2.4.5 Hydrologic analysis must be improved: it is too brief, and the hydrologic tools should be explained in a better way.
Section 3, 4 and 5 are clear and well written.
In the Conclusions section, the further insights of the research should be better investigated.
Author Response
Response to Reviewer 1 Comments
Point 1: In the Abstract it is advisable to add a brief description of the methodological approach used in the research and the main results obtained.
Response 1: Thanks for your kind suggestion. Just like you said, the objective, methods and main results are not very clear in original manuscript. We rewrote the abstract and the detailed revision can be found in revised manuscript, from Lines 13-33, Page 1.
Point 2: In order to improve the Introduction section, it is recommended to add a small paragraph with the explanation of the contents of the subsequent sections.
Response 2: Thanks for your kind suggestion. Just like you said, Introduction section should has explanation of the contents of the subsequent sections. We added explanation of the contents of the subsequent sections and the detailed revision can be found in revised manuscript, from Lines 114-123, Page 3.
Point 3: In section 2 “Materials and Methods” the Figure 3 should be modified. In fact, it is very full of information and not very clear. It is suggested to use the steps of the process (from 1 to 3) as milestones for the graph and to include different forms for different meaning (for example why the years are important? and what is their meaning in the chart?). At least, the verification of the flow of all the arrows should be carried out: it must have a logic sense.
Response 3: Thanks for your kind suggestion. This is a very important suggestion, which would affect the readability of paper. Just like you said, the research approach is full of information, but not very clear in original manuscript. And the reason why we selected the year 2000, 2010, and 2020 is not showed. We reconsidered the logic of research approach and redrew the Figure 3, the detailed revision can be found in revised manuscript, from Lines 160-179, Page 4.
Point 4: In the subsection 2.3, it is advisable to add some information about the data collected and about any operation has been performed in order to test the statistical robustness of the database.
Response 4: Thanks for your kind suggestion. The detailed revision can be found in revised manuscript, from Lines 199-204, Page 5.
Point 5: It is also important to check if all the terms used in the equations are properly introduced and defined.
Response 5: Thanks for your kind suggestion. All the terms used in the equations is cited from published articles, and I have checked all the terms used in the equations.
Point 6: Section 2.4.5 Hydrologic analysis must be improved: it is too brief, and the hydrologic tools should be explained in a better way.
Response 6: Thanks for your kind suggestion. We rewrote the Hydrologic analysis part and the detailed revision can be found in revised manuscript, from Lines 311-322, Page 8.
Point 7: Section 3, 4 and 5 are clear and well written.
Response 7: Thanks for your affirmation and encouragement, I will continue to work hard in future research.
Point 8: In the Conclusions section, the further insights of the research should be better investigated.
Response 8: Thanks for your kind suggestion. This is another very important suggestion. Just like you said, conclusion is lack of usefulness for the field of research and futher insights of the research. After reviewing the limitations of our research, we found that the limitation could help us to do more in-depth research. We rewrote part of conclusion and the detailed revision can be found in revised manuscript, from Lines 662-677, Page 20.

Reviewer 2 Report
The paper “Nature-based solutions in forest-wetland spatial planning strategies to promote sustainable city development in Tianjin, China” focused on two ecological resources “forests and wetlands” to study the synergy and potential trade-offs between them. After reviewing it, I have comments and suggestions as follows.
1). In the abstract, the objective and methods are not clearly presented.
2). Sustainable city development is also a good keyword for your article.
3). I suggest explaining your research approach in more detail and reasonably before its summary in Figure 3. For example, as you mentioned, your study is centered on the process and core thesis of NbS, and it entails: (1), (2) & (3). Why are these three important? Also, the figure shows the year 2000, 2010, and 2020. I did not see any explanation regarding these selected years.
4). I suggest explaining why those formulas in section 2.4 are significant to use.
5). I suggest discussing your results by reflecting/comparing relevant existing literature, avoiding discussing your results alone, especially from lines 411 to 490.
6). The current conclusion is generally like a summary. I suggest making it shorter, stronger, and straight to the point. The conclusion in a scientific article should describe the usefulness of the results in the field of research. Also, please clearly indicate the limitations of your research in the conclusion.
7). Finally, I suggest revising your title and objective to reflect each other. The current title does not properly represent what you have discussed in the paper, and the objective is not clearly presented.
Author Response
Response to Reviewer 2 Comments
Point 1: In the abstract, the objective and methods are not clearly presented.
Response 1: Thanks for your kind suggestion. Just like you said, the objective, methods and main results are not very clear in original manuscript. We rewrote the abstract and the detailed revision can be found in revised manuscript, from Lines 13-33, Page 1.
Point 2: Sustainable city development is also a good keyword for your article.
Response 2: Thanks for your kind suggestion. We added the new keywod and the detailed revision can be found in revised manuscript, from Lines 48-49, Page 2.
Point 3: I suggest explaining your research approach in more detail and reasonably before its summary in Figure 3. For example, as you mentioned, your study is centered on the process and core thesis of NbS, and it entails: (1), (2) & (3). Why are these three important? Also, the figure shows the year 2000, 2010, and 2020. I did not see any explanation regarding these selected years.
Response 3: Thanks for your kind suggestion. This is a very important suggestion, which would affect the readability of paper. Just like you said, the research approach is full of information, but not very clear in original manuscript. And the reason why we selected the year 2000, 2010, and 2020 is not showed. We reconsidered the logic of research approach and redrew the Figure 3, the detailed revision can be found in revised manuscript, from Lines 160-179, Page 4.
Point 4: I suggest explaining why those formulas in section 2.4 are significant to use.
Response 4: Thanks for your kind suggestion. We added the reason why those formulas in section 2.4 and the detailed revision can be found in revised manuscript, from Lines 218-226, Page 6.
Point 5: I suggest discussing your results by reflecting/comparing relevant existing literature, avoiding discussing your results alone, especially from lines 411 to 490.
Response 5: Thanks for your kind suggestion. To avoid discussing the findings in isolation, research results of others are added to support the ideas in this paper. For details, please see revised manuscript lines 469 (Page 15), 526 (Page 15), 537 (Page 15), and 547 (Page 16).
Point 6: The current conclusion is generally like a summary. I suggest making it shorter, stronger, and straight to the point. The conclusion in a scientific article should describe the usefulness of the results in the field of research. Also, please clearly indicate the limitations of your research in the conclusion.
Response 6: Thanks for your kind suggestion. This is another very important suggestion. Just like you said, conclusion is lack of usefulness for the field of research and futher insights of the research. After reviewing the limitations of our research, we found that the limitation could help us to do more in-depth research. We rewrote some part of conclusion and the detailed revision can be found in revised manuscript, from Lines 662-677, Page 20.
Point 7: Finally, I suggest revising your title and objective to reflect each other. The current title does not properly represent what you have discussed in the paper, and the objective is not clearly presented.
Response 7: Thanks for your kind suggestion. The objectives of the study were further specified as“the purpose of this study was to explore the possibility of extending the NbS of a single variable to two variables for better development of sustainable cities.”

Round 2
Reviewer 2 Report
The authors have addressed my comments.